# A New Machine Learning based Analysis for Improving Satellite Retrieved Atmospheric Composition Data: OMI SO₂ as an Example

Can Li[1,2], Joanna Joiner[2], Fei Liu[2,3], Nickolay A. Krotkov[2], Vitali Fioletov[4], Chris McLinden[4]

[1]Earth System Science Interdisciplinary Center, University of Maryland, College Park, MD 20740, USA
[2]Atmospheric Chemistry and Dynamics Laboratory, NASA Goddard Space Flight Center, Greenbelt, MD 20771, USA
[3]Goddard Earth Sciences Technology and Research (GESTAR) II, Morgan State University, Baltimore, MD 21251, USA
[4]Environment and Climate Change Canada, Toronto, Ontario, Canada

*Correspondence to*: Can Li (can.li@nasa.gov)

**Abstract.** Despite recent progress, satellite retrievals of anthropogenic SO₂ still suffer from relatively low signal-to-noise
ratios. In this study, we demonstrate a new machine learning data analysis method to improve the quality of satellite SO₂
products. In the absence of large ground truth datasets for SO₂, we start from SO₂ slant column densities (SCDs) retrieved
from the Ozone Monitoring Instrument (OMI) using a data-drive, physically based algorithm and calculate the ratio between
the SCD and the root mean square (RMS) of the fitting residuals for each pixel. To build the training data, we select presumably
clean pixels with small SCD/RMS ratios (SRRs) and set their target SCDs to zero. For polluted pixels with relatively large
SRRs, we set the target to the original retrieved SCDs. We then train neural networks (NNs) to reproduce the target SCDs
using predictors including SRRs for individual pixels, solar zenith, viewing zenith and phase angles, scene reflectivity and O₃
column amounts, as well as the monthly mean SRRs. For data analysis, we employ two NNs: 1) one trained daily to produce
analysed SO₂ SCDs for polluted pixels each day and 2) the other trained once every month to produce analysed SCDs for less
polluted pixels for the entire month. Test results for 2005 show that our method can significantly reduce noise and artifacts
over background regions. Over polluted areas, the monthly mean NN analysed and original SCDs generally agree to within
±15%, indicating that our method can retain SO₂ signals in the original retrievals except for large volcanic eruptions. This is
further confirmed by running both the NN analysed and the original SCDs through a top-down emission algorithm to estimate
the annual SO₂ emissions for ~500 anthropogenic sources, with the two datasets yielding similar results. We also explore two
alternative approaches to the NN-based analysis method. In one, we employ a simple linear interpolation model to analyse the
original SCD retrievals. In the other, we develop a PCA-NN algorithm that uses OMI measured radiances, transformed and
dimension-reduced with a principal component analysis (PCA) technique, as inputs to NNs for SO₂ SCD retrievals. While the
linear model and the PCA-NN algorithm can reduce retrieval noise, they both underestimate SO₂ over polluted areas. Overall,
the results presented here demonstrate that our new data analysis method can significantly improve the quality of existing OMI
SO₂ retrievals. The method can potentially be adapted for other sensors and/or species and enhance the value of satellite data
in air quality research and applications.

# 1 Introduction

Sulfur dioxide ($SO_2$) and its oxidation product in the atmosphere, sulfate aerosols, have significant impacts on air quality, visibility, ecosystems, and the weather and climate. For over two decades, spaceborne hyperspectral ultraviolet (UV) instruments (e.g., Eisinger and Burrows, 1998; Krotkov et al., 2006; Nowlan et al., 2011; Theys et al., 2017) have been providing global observations of anthropogenic $SO_2$ sources such as coal-fired power plants, metal smelters, and the oil and gas industry (e.g., Fioletov et al., 2016; McLinden et al., 2016; Zhang et al., 2019). More recently, the quality of satellite $SO_2$ data products has substantially improved thanks to the development of data driven retrieval techniques. In particular, the principal component analysis (PCA) based algorithm (Li et al., 2013; 2017a; 2020) and the covariance-based retrieval algorithm (COBRA, Theys et al., 2021) have helped to reduce the noise and artifacts of $SO_2$ retrievals from several sensors including the Ozone Monitoring Instrument (OMI), Ozone Mapping and Profiler Suite (OMPS), and TROPOspheric Monitoring Instrument (TROPOMI), enabling the detection and quantification of relatively small point sources (e.g., Fioletov et al., 2015, Theys et al., 2021).

Despite these progresses, satellite remote sensing of anthropogenic $SO_2$ remains challenging. The signal of anthropogenic $SO_2$ is relatively weak as compared with volcanic sources. With an atmospheric lifetime of ~1 day (e.g., Lee et al., 2011), $SO_2$ emitted from human activities is also more concentrated in the boundary layer, where the sensitivity of satellite instruments is limited by the low surface albedo, strong Rayleigh scattering, and interferences from $O_3$ absorption in the UV (e.g., Krotkov et al., 2008). As a result, the noise in satellite $SO_2$ retrievals is relatively large even for data driven algorithms. For example, the standard deviation (1-σ noise) of OMI PCA $SO_2$ slant column densities (SCDs) over the remote Pacific is ~0.2-0.3 DU (Dobson Unit, 1 DU = $2.69 \times 10^{16}$ molecules/cm$^2$, Li et al., 2020), far greater than the typical SCDs retrieved outside of the most polluted areas (e.g., Persian Gulf, eastern China, and Norilsk, Russia). The retrieval noise can be reduced by spatially and temporally averaging the data (Krotkov et al., 2008). However, relatively small but noticeable artifacts still exist in monthly or annual mean OMI $SO_2$ (e.g., negative values over arid and semi-arid areas), indicating systematic biases that cannot be averaged out. While there was little drift in the mean OMI $SO_2$ SCDs over remote regions from 2005 to 2019, the retrieval noise grew by ~10% during the same period (Li et al., 2020), presumably due to instrument degradation. With the recent large decreases in $SO_2$ emissions and signals in many regions (e.g., Krotkov et al., 2016; Li et al., 2017b), the increase in retrieval noise makes the analyses and applications of satellite $SO_2$ data even more challenging, especially for long-term monitoring. It is thus imperative to further enhance the quality of satellite $SO_2$ data products.

In recent years, machine learning has emerged as a powerful tool in satellite remote sensing of atmospheric composition. Capable of incorporating large, diverse datasets and modelling complex, nonlinear functions, techniques such as neural networks (NN) and random forests (RF) have been utilized to solve various problems. For instance, a number of studies trained NN or RF models to infer surface concentrations of pollutants from satellite observations, including particulate matter (e.g., Huang et al., 2021; Liu et al., 2019; Zheng et al., 2021), $NO_2$ (e.g., Chan et al., 2021), and $SO_2$ (e.g., Zhang et al. 2022). NNs have also been used to speed up radiative transfer calculations (e.g., Castellanos and da Silva, 2019; Nada et al., 2019)

and to retrieve $O_3$ profiles (e.g., Muller et al., 2003; Xu et al., 2017) and total columns (Muller et al., 2004), isoprene amounts (Wells et al., 2020), and aerosol layer height (Chimot et al., 2017). For $SO_2$, De Santis et al. (2021) demonstrated a NN retrieval algorithm using the operational TROPOMI product for training in their case study of Mt. Etna. Piscini et al. (2014) attempted NN-based $SO_2$ and volcanic ash retrievals using thermal infrared measurements from MODIS (the Moderate Resolution Imaging Spectroradiometer). Hedelt et al. (2019) also developed near-real-time volcanic $SO_2$ height retrievals using the Full-Physics Inverse Learning Machine (FP-ILM) method, a technique later adapted for OMI by Fedkin et al. (2021). While these studies have demonstrated the potential of machine learning for $SO_2$ retrievals, they all focus on volcanic $SO_2$. To our knowledge, so far there have been no published studies demonstrating the use of machine learning techniques for anthropogenic $SO_2$ retrievals.

A major obstacle in developing machine learning retrieval algorithms for anthropogenic $SO_2$ is the lack of high-quality, ground-truth training data. As mentioned above, existing satellite $SO_2$ products provide global coverage, but the signal-to-noise ratios are typically small for anthropogenic sources. Ground air quality monitors generally offer good data quality and long-term measurements, but they do not represent the entire atmospheric column. Aircraft measurements and surface based remote sensing instruments (e.g., MAX-DOAS) have been used to evaluate satellite retrievals, but they are quite sparse. The FP-ILM method circumvents this data availability issue by using a large set of model-simulated synthetic radiance spectra in training. However, the models may not fully represent the various geophysical processes and instrument characteristics that affect satellite measurements. This can lead to substantial errors, and FP-ILM retrievals of volcanic $SO_2$ height are currently limited to satellite pixels with sizable $SO_2$ amounts (> 20 DU).

Here, we introduce a new data analysis method to further improve the quality of satellite retrieved $SO_2$. In the absence of sufficient ground truth data, we compile our training data by analysing existing OMI $SO_2$ SCD retrievals and the associated fitting errors, assuming that retrievals with greater SCDs and smaller fitting errors can be trusted more than those with smaller SCDs and larger errors. This allows us to train NNs to reduce noise and artifacts in the original retrievals, meanwhile retaining $SO_2$ signals over major emission source areas. The rest of the paper is organized as follows: Section 2 describes our methodology and setups for NN training. Section 3 provides some example results. This is followed by a more detailed discussion on the NN analysed SCDs in Section 4 and conclusions in Section 5.

## 2 Data and methodology

The flowchart in Fig. 1a presents an overview of our data analysis method. We start from existing OMI PCA $SO_2$ retrievals (Section 2.1) and calculate the ratio between the SCD and the root mean square (RMS) of the fitting residuals (SCD/RMS ratio, SRR) for each pixel, as well as the statistics of the SRRs for the entire month. This provides input to a data classification scheme (Fig. 1b, Section 2.2) that assigns OMI pixels from each day into different groups ("clean", "polluted", "in-between" and "high-SRR"). The pixels within each group are then either processed with one of the two neural networks (pre-trained NN1 for clean and in-between pixels, daily trained NN2 for polluted pixels, Fig. 1c, Section 2.3) or retain their original

retrieved SCDs (for high-SRR pixels). In the end, the OMI pixels from different groups are merged into the final analysed SCD dataset.

## 2.1 Analysis of OMI SO$_2$ data

To demonstrate our methodology, we use data from OMI, a Dutch/Finnish UV/Visible spectrometer that has been flying on the National Aeronautics and Space Administration (NASA)'s Aura spacecraft in a Sun synchronous orbit since 2004 (Levelt et al., 2018). OMI measures backscattered solar radiation between 270 and 500 nm in the local afternoon (local equator crossing time: ~13:45) at a relatively high spatial (13 × 24 km$^2$ at nadir) and spectral (~0.5 nm) resolution. We focus on the year 2005, when all cross-track positions (rows) of OMI's 2-dimensional detectors were taking nominal measurements, providing daily global coverage.

For SO$_2$ data, we use SCDs retrieved from NASA version 2 OMI standard SO$_2$ algorithm based on the PCA spectral fitting technique. The algorithm has been described in detail elsewhere (Li et al., 2020) and is only briefly introduced here. The algorithm uses OMI-measured Sun-normalized Earthshine radiances within the spectral range of 310.5-340 nm and processes each row of individual OMI orbits separately. The ~1600 OMI pixels from a given row in a given orbit are first filtered to exclude those with large solar zenith angles (SZA > 75º) or potentially strong SO$_2$ signals (e.g., volcanic plumes by examining the ozone residuals at 313/314 and 314/315 nm wavelength pairs, see Li et al., 2017a for details). Next, the spectra of the remaining pixels are analysed utilizing a PCA technique to extract spectral features (principal components, PCs). The leading PCs that account for the most spectral variances are typically associated with geophysical (e.g., O$_3$ absorption and rotational Raman scattering, RRS) or instrumental (e.g., dark current, wavelength shift) factors that interfere with SO$_2$ retrievals. For each pixel, up to 30 leading PCs, along with the SO$_2$ cross sections, are fit to the measured radiances to estimate the SO$_2$ SCD while minimizing the interferences. This multi-step (data filtering, PCA analysis, and spectral fitting) procedure is iterated a few times. To avoid collinearity in fitting, the PCs are also examined to exclude those potentially containing SO$_2$ spectral signatures. For this study, the standard algorithm has been modified to use the new collection 4 OMI level 1B (L1B) radiance and irradiance data, instead of the collection 3 data for the current standard OMI SO$_2$ product. No obvious differences were found between the SCDs retrieved from the two collections. In addition, the RMS of the fitting residuals (i.e., the differences between the measured and the fit normalized radiance spectra) for each pixel has been added to the output.

In order to compare the SO$_2$ signal *vs.* the fitting error, we calculate the SCD/RMS ratio (SRR) for each pixel. The pixel-level SRRs are also gridded into monthly mean (SRR$_m$) at 0.25º × 0.25º resolution (Fig. 2). At middle and low latitudes, the overall spatial distribution of SRR$_m$ (Fig. 2b) is quite similar to that of the monthly mean SCDs (Fig. 2a). On the other hand, the bias in SRR$_m$ is smaller at high latitudes due to generally greater fitting errors at larger SZAs, allowing us to better distinguish polluted areas from background regions. In the following steps (Sections 2.2 and 2.3), SRR$_m$ is utilized as an indicator of the likelihood of an OMI SO$_2$ retrieval over a certain area to represent a positive SO$_2$ value. For each day of the month, we also calculate the mean and standard deviation of SRRs for 3º latitude bands, using all pixels within each band after

removing outliers (SRRs outside of $\pm5\sigma$ from the mean). The monthly medians of the daily mean ($\overline{SRR}$) and standard deviation ($\sigma_{SRR}$) are then taken from each latitude band as inputs to the OMI pixel classification scheme (Section 2.2)

## 2.2 Classification of OMI pixels

The purpose of the pixel classification scheme (Fig. 1b) is to compile a training dataset by selecting pixels in two categories: 1) the first for clean pixels in which the retrieved SCDs are relatively small while the fitting errors are relatively large (i.e., negative or small positive SRRs) so that they can be considered largely $SO_2$-free and 2) the second in which the retrieved SCDs are large while the fitting errors are relatively small (i.e., large SRRs). In this category for polluted pixels, the retrieved SCDs are assumed to be close to the truth. There are two additional categories. The third is for pixels that fall in-between the clean and polluted categories. For these pixels, an unambiguous classification cannot be made and they are excluded from the training dataset. The fourth category (high SRR) is for pixels that have very large SRRs (> 300). Such pixels are few but are also excluded from the training, as they tend to have a disproportionally large influence on the trained NNs.

In addition to the SRRs of individual OMI pixels, the classification scheme also takes into account the location (latitude/longitude) of the pixels, as well as the general performance of the PCA algorithm for the latitude bands in which they are located. A pixel with a specific SCD/RMS ratio of $SRR_i$ is considered to be polluted, if:

$$SRR_i > \overline{SRR} + a_1\,\sigma_{SRR}\;. \tag{1}$$

The pixel would be considered to be clean, if:

$$SRR_i < \overline{SRR} + a_2\,\sigma_{SRR}\;, \tag{2}$$

where $\overline{SRR}$ and $\sigma_{SRR}$ are the monthly medians of the daily mean and standard deviation of SRRs for the corresponding latitude band, respectively. $a_1$ and $a_2$ are scaling factors (see Fig. 1b for values) that have been adjusted through trial and error in order to 1) minimize the artifacts in NN analysed SCDs over background areas and 2) maximize the retained original $SO_2$ signals over polluted areas. Both factors depend on the location of the pixels and the monthly mean SRRs ($SRR_m$). As shown in Fig. 1b, $a_1$ and $a_2$ are large, if the pixel is located in an area with a small $SRR_m$ (< 3). In this case, the area is generally unpolluted and the likelihood of a pixel containing a positive $SO_2$ value is low. Thus, more pixels are classified as clean. On the other hand, for polluted areas with large $SRR_m$ (> 5), both $a_1$ and $a_2$ are kept small so that more pixels would be classified as polluted. For areas that are moderately polluted (i.e., $3 < SRR_m < 5$), $a_1$ and $a_2$ are linearly interpolated based on the $SRR_m$. One may also notice that $a_1$ and $a_2$ are smaller for low (30ºS-30ºN) and middle (30ºS-60ºS and 30ºN-60ºN) latitudes than for high latitudes. This helps to reduce the positive bias in the original SCDs near the northern edge of the domain (Fig. 2a). We also tested a simple classification scheme with constant $a_1$ and $a_2$ everywhere, and found that it produces relatively large positive biases over high latitudes and negative biases over low latitude source areas, as compared with the more complicated scheme described above. It should also be pointed out that the areas affected by the south Atlantic anomaly (SAA) are not subject to classification and excluded from the training dataset.

Using the classification scheme, one can also develop a simple method to reduce retrieval artifacts, by assuming that clean pixels should have zero SCDs while polluted pixels should retain their original SCDs and by estimating SCDs for pixels

that fall in-between through a linear interpolation (between zero and the original SCDs). As will be demonstrated in Section 4.4, such an approach produces negative biases over polluted areas. It is thus advantageous to employ the more complex NN-based method for the present study.

### 2.3 Training of neural networks

For training data, we use the OMI pixels identified as either clean or polluted by the classification scheme. For a typical day, approximately ~800000 out of ~1 million OMI pixels are classified as clean, and ~10000 (~1%) as polluted. Given the scarcity of ground truth $SO_2$ data, we set the training target ($SCD_{target}$) to zero for the clean pixels and to the original SCDs for the polluted pixels. Note that unlike the PCA spectral fitting algorithm, data from all 60 rows are pooled together in the training so that a relatively large sample of polluted pixels is available. We also include several candidate predicators in the training data, including SCD/RMS ratios for the individual pixels ($SRR_i$), the monthly mean SCD/RMS ratios ($SRR_m$) where the pixels are located, the cosines of solar zenith angles (SZA, $\theta_0$), viewing zenith angles ($\theta$) and phase angles ($\phi$), the $O_3$ column amounts from the OMI total $O_3$ product (OMTO3, Bhartia, 2005), and the scene reflectivity ($R$) at 354 nm from the OMI Raman cloud product (OMCLDRR, Joiner and Vasilkov, 2006). The function of a neural network (NN) is then to use the input predictors or features to predict the output $SCD_{target}$:

$$SCD_{target} = f_{NN}(SRR_i, SRR_m, \theta_0, \theta, \phi, R, O_3) . \tag{3}$$

To optimize the set of predictors, we carried out a number of tests using different combinations (See Table 1 for example results). Among the predictors, $SRR_i$ is well correlated with $SCD_{target}$ and has the largest impact on the performance of the NNs. Indeed, the NN without $SRR_i$ produces the lowest correlation coefficient ($r$) and the largest root mean square error (RMSE) between the analysed SCDs ($SCD_{NN}$) and $SCD_{target}$ (Table 1). $SRR_m$ provides geospatial context for the NNs so that higher SCDs tend to be assigned to polluted areas. In the particular example in Table 1, a simple NN using just $SRR_i$ and $SRR_m$ as predictors produces $SCD_{NN}$ that agrees reasonably well with $SCD_{target}$ ($r = 0.958$, RMSE = 0.0517 DU). The angles, $O_3$ column amounts, and scene reflectivity all affect the signal-to-noise ratio of OMI measurements and the quality of $SO_2$ retrievals (Li et al., 2020). Adding them as predictors generally leads to small but noticeable improvements in the performance of the NNs (Table 1). While the NN with all seven predictors has slightly worse performance than the NN without $SRR_m$ for this case, including $SRR_m$ as a predictor helps to retain signals over $SO_2$ source areas. We also tested additional predictors (e.g., the terrain pressure and the scene pressure) and found no discernible improvements in the overall performance of the NNs. Hereafter we use all seven predictors as specified in Eq. 3 in the NNs.

The architecture of the NNs in this study (Fig. 1c) is similar to that employed by Joiner et al. (2021, 2022) for reconstruction of RGB images from hyperspectral radiances. A similar architecture has also been used to capture changes in gross primary production (GPP) from satellite reflectance data (Joiner and Yoshida, 2020). Briefly, the artificial feedforward NNs are implemented in IDL (Interactive Data Language) and contain two hidden layers, each with 14 nodes (twice the number of predictors), and an output layer with one node. Experiments using more (up to 30) nodes in each hidden layer yield little difference in the performance of the NNs. The activation functions are a soft sign for the first hidden layer, a logistic (sigmoid)

for the second hidden layer, and a bent identity for the output layer. If we replace the activation functions in both hidden layers
with ReLU (Rectified Linear Unit), the NNs converge faster in training but increase the SCDs over background areas by ~0.01-
0.02 DU (Fig. S1). An adaptive moment estimation (Adam) optimizer (Kingma and Ba, 2014) with a learning rate of 0.1 is
used to minimize the error. Inputs and outputs are normalized so that they each have a mean of zero and a unit standard
deviation.

For each month, we train a neural network (NN1, Fig. 1) utilizing data from 5 days (the 5th, 10th, 15th, 20th and 25th
200    days of the month). Half of the clean and polluted pixels are used in the training and the rest for evaluation. We notice that
NN1 well reproduces $SCD_{target}$ for clean pixels and also for polluted pixels that have SCDs up to ~4-5 DU, but it produces a
low bias for larger SCDs. This is likely due to the imbalance between the clean and polluted categories in the training data. To
mitigate this issue, we use the pre-trained NN1 only for clean and in-between pixels (Fig. 1a) and a separate neural network
(NN2) for polluted pixels from each day (Fig. 1a). NN2 has the same architecture as NN1 but is trained daily with half of the
polluted pixels. Alternatively, we can also train NN2 using data from multiple days and apply the pre-trained multi-day model
to the entire month. As compared with the daily trained NN2, $SCD_{NN}$ produced by the multi-day model is similar but slightly
lower over some polluted areas (e.g., eastern China). To maximize the retained $SO_2$ signals over those regions, we use daily
trained NN2 in the present study.

In the final step (Fig. 1a), the $SCD_{NN}$ outputs from NN1 and NN2 are merged with the original SCDs for high-SRR
pixels to produce the final NN analysed SCDs.

## 3 Results

### 3.1 Daily comparisons of $SO_2$ SCDs

In Fig. 3, we compare the NN analysed $SO_2$ SCDs ($SCD_{NN}$) and the target SCDs ($SCD_{target}$) from the 16th of January, April,
July and October 2005, for independent pixels that are not part of the training. There is generally good agreement between
$SCD_{NN}$ and $SCD_{target}$, with $r > 0.93$ and RMSE at ~0.02-0.03 DU for all four days. The vast majority of clean pixels as identified
by the classification scheme have $SCD_{NN}$ between -0.1 and 0.1 DU, indicating substantial reduction in the retrieval noise as
compared with the original retrievals (1-$\sigma$ noise of ~0.2-0.3 DU), although a small fraction of the clean pixels still have $SCD_{NN}$
as large as ±0.5 DU. The slopes from the linear regression analysis are between 0.95 and 0.98, suggesting slight underestimates
in $SCD_{NN}$. There is also some scatter for the polluted pixels particularly at higher SCDs (> 2 DU). The number of pixels having
large $SCD_{target}$ are relatively small and this limit in the training data may affect the performance of NNs under high SCD
conditions (such as for volcanic plumes). We repeated the analysis for the whole year and found similar results for most days.
On average, the correlation coefficient from the daily comparisons is 0.948 ± 0.0309 (hereafter results are shown as mean ±
standard deviation), the RMSE is 0.0343 ± 0.0194 DU, while the slope is 0.966 ± 0.0409. There are four days with RMSE >
0.1 DU (April 6, June 11, July 13, and August 14). All four have relatively large errors over areas affected by volcanic plumes,

again suggesting that the NN performance may deteriorate at high SCDs. Overall, the comparisons here point to quite good

performance of the NNs in reproducing the target SCDs.

As compared with the original SCDs, the NN analysed SCDs have much reduced noise and artifacts over background areas and largely retain $SO_2$ signals over polluted regions. This is evident from Fig. 4 which shows the original $SO_2$ SCDs, the NN analysed SCDs, their differences, and the mean SCDs for different latitude bands over generally clean areas (monthly

mean SRR < 3) for April 16, 2005 as an example. As can be seen from the figure, the NN analysed SCDs show little variation with latitudes as compared with the original PCA retrievals (Fig. 4d). The differences between the two (Fig. 4c) are similar to the original SCDs (Fig. 4a) over most background areas, as ~80% of the pixels are identified as clean and have $SCD_{NN}$ within ±0.1 DU. The differences are quite small over polluted regions (e.g., eastern China, Sichuan Basin, Norilsk), as pixels over those areas tend to be classified as polluted and have $SCD_{NN}$ close to their original retrievals. It is worth mentioning that even

though the SAA affected areas are excluded from training, the analysed SCDs over those areas still show smaller noise than the original ones. One potential reason is that retrievals over the SAA areas tend to have relatively large RMS, and the use of SRRs partially cancels out the relatively noisy SCDs.

**3.2 Comparisons of monthly $SO_2$ SCDs**

The monthly maps in Fig. 5 for March 2005 show consistent results with the daily comparisons in Section 3.1. While the

monthly mean SCDs from the original PCA retrievals (Fig. 5a) are close to zero for most background areas, biases are evident over certain regions. For example, there are patches of negative SCDs (approximately -0.1 DU) at ~40-60ºN and over the oceans near the equator. Another noticeable feature is the negative bias over the relatively bright arid and semi-arid land surfaces such as the Sahara desert, the Arabian peninsula, and the Taklimakan and Gobi deserts. It is possible that the retained PCs (derived from hundreds of pixels from each OMI row) do not fully capture certain interfering factors for those areas. The

exact reasons for these artifacts are unknown and beyond the scope of the present study. In any case, they are largely removed through our NN-based analysis (Fig. 5b). Meanwhile, there is no obvious difference between the original and analysed SCDs over major $SO_2$ source areas (Fig. 5c), evidence that the NNs have learned to preserve the $SO_2$ signals over those areas.

One may notice that outside of the source regions, the difference map in Fig. 5c is not identical to the original SCD map in Fig. 5a. For example, the differences are slightly more negative than the original SCDs over parts of Canada, Mongolia,

and Russia. Most pixels have $SCD_{NN}$ near zero, but some pixels with noisy, positive original SCDs could be misclassified as polluted, resulting in a small positive bias in $SCD_{NN}$ for these areas. This is also noticeable in Fig. 5d that shows the mean original and NN analysed SCDs within 1º latitude bands over clean areas. The NN SCDs have generally less structure, indicating reduced artifacts, but a positive bias of ~0.02 DU can be found north of 60ºN. Mean SCD maps for other months (January, April, July, October 2005, see Fig. S2 in the supplemental information) show quite similar results. For areas/periods

strongly influenced by relatively large volcanic eruptions (e.g., Sierra Negra (Galapagos Islands) eruption in October 2005), the NNs have difficulty completely reproducing the strong $SO_2$ signals. This again points to the slightly deteriorated performance of NNs under high $SO_2$ conditions, as already discussed.

A close-up look at the NN-analysed SCDs and their differences from the original SCDs over eastern China is given in Fig. 6. For polluted areas (analysed SCDs > 0.15 DU), the relative differences are typically within ±20%, with a mean of 4% (with the original SCDs being greater). For background areas, the relative differences are close to ±100% as expected for clean pixels. Comparisons for other major anthropogenic source areas including India, the Middle East, South Africa, the eastern U.S., and Norilsk, Russia yield similar results (see Figs. S3-S7 in the supplemental information). The mean relative differences for polluted areas in these regions are all within ±15%, ranging between -11% for the eastern U.S. and 14% for the Middle East. In comparison, the relative differences for areas affected by large volcanic plumes are greater, for example reaching 20% on average over part of the southeast Pacific during the October 2005 Sierra Negra eruption (see Fig. S8 in the supplemental information).

## 4 Discussion

### 4.1 Original and analysed $SO_2$ SCDs as a function of SRRs

The results presented in Section 3 demonstrate that our NN-based analysis can reduce noise and artifacts for clean pixels, meanwhile largely retaining the original SCDs for polluted pixels. However, some key questions remain unanswered. Namely, given the somewhat subjective criteria used in the pixel classification scheme (Section 2.2) to build the training data, do we risk removing real $SO_2$ signals as noise (i.e., over-correction) and/or keeping noise/artifacts as signals (i.e., under-correction)? Another related question is: how do the NNs treat pixels that are not in the training data (i.e., the pixels that fall in-between the clean and polluted categories)? To shed light on these issues, we calculate the monthly mean $SO_2$ SCDs as a function of pixel-level SCD/RMS ratios ($SRR_i$) from the original retrievals (Panel a of Figs. S9-S13 in the supplemental information) and the analysed data (Panel b of Figs. S9-S13 in the supplemental information), as well as their differences (Fig. 7, left) for March 2005. In addition, we also calculate the mean original and NN analysed SCDs as a function of latitude for different ranges of $SRR_i$ (Fig. 7, right).

For pixels having $SRR_i < \overline{SRR}$ (see Section 2.1 for definitions of $\overline{SRR}$ and $\sigma_{SRR}$), the original SCD map (Fig. S9a) shows no obvious hotspots even over the major $SO_2$ source areas. All such pixels would be classified as clean and indeed the mean NN analysed SCDs (Fig. S9b) from these pixels are zero everywhere. The mean analysed SCDs (Fig. 7b) are also near zero at all latitudes.

The next group of pixels have $\overline{SRR} < SRR_i < \overline{SRR} + \sigma_{SRR}$ (Fig. 7, second row). Most pixels in this group, except for those near large $SO_2$ sources at low latitudes (30ºS-30ºN), would also be classified as clean. Similar to the first group, there are no obvious $SO_2$ hotspots in the original SCD map (Fig. S10a). The analysed SCDs (Fig. S10b) are similarly near zero almost everywhere, with notable exceptions over some degassing volcanoes (Anatahan, Nyiragongo, and Vanuatu) and heavily polluted areas (Sichuan Basin and Norilsk). The case of Norilsk is particularly interesting. Given the thresholds for high latitudes (Fig. 1b), all pixels in this group over Norilsk would be classified as clean, but the NNs seem to be able to override

the classification based on factors other than SRRs. The mean analysed SCDs are around zero for all latitude bands, smaller

than the original SCDs (Fig. 7d).

For the group of pixels having intermediate SRRs ($\overline{SRR} + \sigma_{SRR} < SRR_i < \overline{SRR} + 2\sigma_{SRR}$, Fig. 7, third row), the original SCD map (Fig. S11a) contains enhanced $SO_2$ signals over source areas but also artifacts over background regions. The pixels in this group would be classified as clean, polluted, or in-between depending on their $SRR_i$ and locations. In general, the NNs are able to largely eliminate the artifacts and retain signals over $SO_2$ source areas for this group (Fig. 7e), although there are

remaining small positive biases both near the northern edge of the domain and at lower latitudes (e.g., around 30ºS) as shown in Fig. 7f.

For the following group ($\overline{SRR} + 2\sigma_{SRR} < SRR_i < \overline{SRR} + 3\sigma_{SRR}$, Fig. 7, fourth row), almost all pixels would have a classification of either polluted or in-between. NNs reduce the retrieval artifacts in this group particularly at middle to high latitudes (Figs. 7g and 7h). The relatively small changes at low latitudes can probably be attributed to the more relaxed

thresholds for pixels to be classified as polluted and in-between (Section 2.2). Using more stringent thresholds may further reduce the artifacts in the tropics, but this may also lead to low bias over pollution sources (See Fig. 8b for example).

For the final group ($SRR_i > \overline{SRR} + 3\sigma_{SRR}$, Fig. 7, fifth row), almost all pixels are identified as polluted. As a result, the differences between the original and the analysed SCDs are quite small except over the SAA affected areas (Figs. 7i and 7j at around 30ºS) where the pixels are not part of the training data and the noise is reduced by the NNs. Overall, it is

encouraging that the NN analysed SCDs show improvements over the original ones for all ranges of the SRRs.

## 4.2 Sensitivity of NN analysed SCDs to the pixel classification scheme

We further test the sensitivity of NN analysed SCDs to the settings of the pixel classification scheme, by altering the $a_1$ and $a_2$ parameters (Eq. 1 and 2). In one experiment, we scale both parameters by 90% (i.e., reduced by 10% from the baseline values as specified in Fig. 1b). This leads to more pixels being classified as polluted and greater monthly mean SCDs (Fig. 8a). The

increase in the SCDs is ~0.01-0.02 DU on average over relatively clean areas (Fig. 8c) and slightly larger over some source areas (e.g., eastern China) but typically less than 0.1 DU. In another experiment, both $a_1$ and $a_2$ are scaled by 110% (i.e., increased by 10% from the baseline values), resulting in SCD reductions of ~0.01 DU over clean areas (Fig. 8c). Some source areas (e.g., Norilsk) show slightly more reductions (Fig. 8b) that are still typically less than 0.1 DU. Overall, the tests here point to moderate sensitivity of the NN-based analysis to the settings of the pixel classification scheme. An overly stringent

scheme may lead to over-correction and low biases over source areas, whereas an overly relaxed scheme may result in positive biases. For our particular study, any over- or under-correction appears to be minor for major source areas, given the relatively small differences between the analysed and the original SCDs (see Sect. 3.2). But if one is to apply the technique to other datasets (e.g., different instruments and/or species), the pixel classification scheme will need to be tested and optimized. For long-term analysis of a dataset from a single instrument (e.g., OMI $SO_2$ for the entire mission), the scheme will need to be

verified using data from different years, although we envision that a constant set of $a_1$ and $a_2$ parameters over time will probably

be more suitable to avoid artificial trends introduced by time-dependent parameters. For future studies, we plan to develop a more systematic way for pixel classification, for example, by using more objective metrics.

## 4.3 SO₂ emission estimates using the original and NN analysed SCDs

Another test involves running both the original and NN analysed SCDs through a top-down emission estimation algorithm to derive annual $SO_2$ emissions from large point sources. Here we focus on anthropogenic sources, given the low bias in the NN analysed SCDs for large volcanic plumes. We infer $SO_2$ emissions by fitting oversampled and smoothed OMI vertical column densities (VCDs) to a 3-parameter (i.e., total mass, lifetime and plume spread) function of horizontal coordinates and wind speeds (Fioletov et al., 2015). To convert SCDs to VCDs, we use the same air mass factors (AMFs, VCD = SCD/AMF) as in Fioletov et al. (2016). For wind fields, we use the average winds between the surface and ~1 km from GEOS-5 Forward Processing for Instrument Teams (FP-IT) assimilated products that have been co-located with OMI (OMUFPITMET; available at https://disc.gsfc.nasa.gov/datasets/OMUFPITMET_003/summary). The OMI pixels are then rotated around known source locations according to wind directions such that all observations are aligned in the upwind-downwind direction (Fioletov et al., 2015). Following Fioletov et al. (2016), we prescribe the $SO_2$ lifetime (6 h) and the parameter describing the spread of the emitted plume (20 km) to obtain more robust fitting results. Only one parameter, the total $SO_2$ mass, is estimated from the fit. We further derive $SO_2$ emissions by dividing the fitted total $SO_2$ mass by the prescribed lifetime. For fitting uncertainty, we calculate the one standard deviation error in the fitted parameter by taking the square root of the diagonal elements of the covariance matrix of the parameter.

As shown in Fig. 9a, the two sets of emission estimates agree quite well ($r > 0.99$, slope $> 0.96$), suggesting the NN analysis has largely preserved $SO_2$ signals in the original retrievals. In general, the estimated emissions using the NN analysed SCDs are slightly smaller than those based on the original retrievals, particularly for relatively small sources ($< 20$ kt, $10^3$ tonnes, per year). While on the surface this may suggest loss of some real $SO_2$ signals in our analysis for relatively small sources, the emission uncertainties (Fig. 9b) for those sources also become much smaller when using the NN analysed data. This leads to greater emission/uncertainty ratios (Fig. 9c) for those sources, implying that the reduced noise/artifacts in the analysed data may facilitate $SO_2$ source detection and quantification. We note that the results here should be interpreted with caution, given that OMI sensitivity to sources $< 30$ kt/year is quite limited (Fioletov et al., 2015).

## 4.4 Can a simple linear interpolation model reproduce NN analysed SCDs?

Given the seemingly simple assumptions made about the clean and polluted pixels during the training process (Section 2.3), one may also ask whether there is any advantage to using the NN-based data analysis approach. To test this, we apply the same pixel classification scheme as described in Section 2.2 and build a simple model by assigning zero SCDs to the clean pixels, the original SCDs to the polluted pixels, and by linearly interpolating between zero and the original SCDs for pixels that fall in-between (based on the SRRs for those pixels and the corresponding thresholds as defined in Eq. 1 and Eq. 2).

The mean SCDs for March 2005 (Fig. 10a), produced with this simple linear interpolation model, appear to be quite similar to those produced with the NN-based analysis (Fig. 5b). This is not surprising since the majority of pixels are classified as clean, and NN analysed SCDs for those pixels are also close to zero. The plot of mean SCDs as a function of latitude (Fig. 10c) also indicates overall comparable results for relatively clean areas between the two methods, although the linear model has more obvious step changes at 30ºN and 30ºS probably related to the pixel classification scheme. Over pollution source areas (e.g., eastern China), on the other hand, the linear model has a substantial negative bias as compared with the NN-based approach (see the SCD difference map in Fig. 10b). Additionally, the noise is also larger over the SAA areas for the linear model. This comparison demonstrates some advantages in the NN-based approach, particularly for preserving $SO_2$ signals over source areas. It should be mentioned that the simple linear model tested here can be potentially improved by including more predictors such as those used in the NNs (e.g., monthly SRRs, the Sun-satellite geometry, and $O_3$). But such a multi-regression model may need to be optimized locally for different regions and can be more challenging to implement, as compared with the NNs.

## 4.5 Implementation of a PCA-NN $SO_2$ fitting algorithm

So far, we have relied on the output from the existing PCA $SO_2$ algorithm as input to the NNs; therefore, our method can be viewed as an additional data processing step following the spectral fit. For a potential alternative to this approach, we also attempt to build an NN-based $SO_2$ SCD fitting algorithm that uses the measured radiances as inputs and the NN analysed SCDs for training targets. As with the PCA $SO_2$ algorithm, the NN fitting algorithm uses the logarithm of Sun-normalized Earthshine radiances at 310.5-340 nm and processes each OMI row separately with individually trained NNs. We pool the data from 12 days in 2005 (the 10th day of each month), generating a training dataset that contains about 200000 pixels for each row. To reduce the data dimension of the inputs, a PCA technique is combined with the NNs in this PCA-NN fitting algorithm as in Joiner et al. (2022). We conduct PCA on the radiance spectra and include the coefficients of the first 50 leading PCs as predictors in the NNs. Experiments using fewer (as few as 20) or more (up to 100) PCs generally result in larger errors in the retrieved SCDs. In addition to the PC coefficients, the NNs also use four other parameters (solar zenith angles, $O_3$ column amounts, scene reflectivity, and monthly mean SRR ratios) as predictors. Viewing zenith angles are not included since the training is carried out separately for each row. We also exclude the phase angles, given that adding them as a predictor leads to no discernible improvements in the algorithm performance. The SRRs for individual pixels are also excluded, as the PCA-NN algorithm is designed to run independently from the PCA $SO_2$ algorithm after the training phase. While the monthly mean SRRs also originate from the PCA retrievals, they essentially provide geospatial context on the spatial distribution of $SO_2$ and can be potentially replaced with other datasets such as $SO_2$ emission inventories or model simulated $SO_2$.

For the PCA-NN algorithm, we utilize a NN architecture similar to that in Fig. 1c, with the only difference being that the number of nodes in each hidden layer is now 108 (twice the number of the predictors). For each row, we train an NN using half of the pixels and the rest for evaluation. The pre-trained NNs are then applied to $SO_2$ SCD retrievals for April 16, 2005, a day not used in the training.

The results shown in Fig. 11 indicate that the PCA-NN algorithm can reduce the retrieval noise over background areas as compared with the original PCA SO$_2$ algorithm. However, over polluted areas and degassing volcanoes, the PCA-NN retrieved SO$_2$ is biased low (Fig. 11c). This suggests that the PCA-NN algorithm, with its present implementation, cannot yet achieve the same level of performance as our NN-based data analysis on the original PCA retrievals. It is possible that due to the much smaller number of polluted pixels as compared with the clean ones, some spectral signatures of SO$_2$ are lost in the

first 50 or even 100 PCs, leading to the low bias over polluted areas. The NNs may need to include more PCs as predictors or directly use radiances without the transformation. A separate set of NNs trained on a refined dataset that contains more polluted pixels may also help to mitigate the bias. But applying these NNs to retrievals would require some prior knowledge about the status of the pixels (whether they are polluted or clean). Also, the PCA-NN retrievals show some striping features, probably reflecting the different performance of the NNs for different rows despite the use of the same architecture. The reason for this

row-to-row change in performance is not yet understood. Nonetheless, the PCA-NN algorithm shows promises and will be the subject of more in-depth studies in the future. For example, the training performance may improve if the architecture is optimized for each row.

## 5 Conclusions

We have developed a new machine learning based method to analyse satellite retrieved atmospheric composition data, with

the aim to reduce the noise and artifacts while retaining the signals in the original retrievals. To demonstrate this approach, we use OMI SO$_2$ SCDs retrieved with the PCA-based spectral fitting algorithm as an example. A key parameter in the analysis method is the SRR, the ratio between the retrieved SCD and the RMS of the fitting residuals. Based on prior knowledge about the global distribution of SO$_2$ pollution (from existing in situ measurements and model simulations), we assume that a given pixel with a small (large) SRR is likely clean (polluted) and its real SCD should be close to zero (the original retrieved SCD).

This allows us to overcome the lack of ground truth data and build a training dataset for SO$_2$ by selecting clean and polluted pixels from the original retrievals.

      We then train neural networks (NNs) using the compiled dataset. The NNs contain two hidden layers with 14 nodes each and one node in the output layer for the analysed SCDs. The predictors for the NNs include SRRs for individual pixels, solar zenith, viewing zenith and phase angles, scene reflectivity, and O$_3$ column amounts, as well as the monthly mean SRRs.

The latter provide context for the spatial distribution of SO$_2$, whereas the other predictors (angles, O$_3$ and reflectivity) affect the quality of the original SCDs. The function of the NNs is to connect these predictors to the target SCDs (zero for clean pixels, the original SCDs for polluted pixels in the training data). For data analysis, we employ a hybrid model (Fig. 1) that includes two NNs: 1) an NN pre-trained using 5 days of data from each month to produce analysed SO$_2$ SCDs for pixels that are clean or moderately polluted (i.e., those with SRRs in between clean and polluted pixels) for the entire month and 2) an

NN trained daily to produce analysed SCDs for the polluted pixels each day. This hybrid model helps to maximize the retained SO$_2$ signals over source areas.

Results for 2005 show that the NNs can well reproduce the target SCDs and largely reduce noise and artifacts in the original retrievals. For polluted areas, the monthly mean SCDs from the analysis are mostly within ±15% from the original retrievals, indicating that the NNs are able to preserve $SO_2$ signals. This is confirmed by another experiment in which the NN

analysed and original SCDs are used to estimate the $SO_2$ emissions for ~500 anthropogenic sources in 2005, with both datasets yielding largely similar results. For relatively small sources (< 20 kt/year), the emission estimates based on the analysed SCDs are generally smaller, but the uncertainties for those sources are reduced even more, although OMI has quite limited sensitivity to such small sources. One remaining issue is that the NNs perform slightly worse for high $SO_2$ conditions such as plumes from large volcanic eruptions (e.g., the 2005 Sierra Negra eruption). This will be the focus of future studies to further improve

the method. Also the NNs analysed SCDs show moderate sensitivity to the settings of the pixel classification scheme. Therefore the scheme needs to be tested, especially for different instruments and/or species, to minimize over- or under-correction. Overall, it is quite encouraging that the NNs seem to have improved the quality of SCDs for pixels from different ranges of SRRs.

We also compare two alternative approaches with the NN-based analysis method. In one test, we employ a simple

linear interpolation model to analyse the original retrievals. The linear model can largely match the performance of NNs over background areas, but underestimates $SO_2$ over polluted regions. In another test, we develop a PCA-NN algorithm that first transforms OMI measured radiances using a PCA technique and then uses the resulting PC coefficients as predictors in NNs (trained with NN analysed SCDs) for $SO_2$ retrievals. Again, the PCA-NN algorithm can reduce retrieval noise but also has a low bias over $SO_2$ source areas. One advantage of the PCA-NN algorithm is its computation speed (approximately a factor of

two faster than the original PCA algorithm in our limited tests) that can make it useful for high resolution instruments such as TROPOMI or TEMPO (Tropospheric Emissions: Monitoring of Pollution). Further improvement in the PCA-NN $SO_2$ algorithm may be possible through, for example, refinement of the training data and will be the subject for follow-up studies. The lack of the high-quality training data has been a major obstacle for training NNs to conduct retrievals using radiances (or PCA transformed radiances). Our analysis method can contribute to such efforts by providing training data with improved

quality as compared with the original retrievals.

In summary, our new machine learning based data analysis method shows promises in further improving satellite retrievals of atmospheric composition. In a way, our analysis method can be viewed as a more advanced version of the Pacific sector correction (PSC), a quite common and well-established practice to reduce retrieval artifacts for species such as $SO_2$ (e.g., Theys et al., 2017). While we focus on OMI $SO_2$ in this study, the method can also be potentially applied to other

instruments (e.g., TROPOMI) and/or species (e.g., HCHO). The improved data quality from such analyses will likely enhance the value of satellite data in air quality research and applications such as reducing the uncertainty in top-down emission estimates.

## Code and data availability

Collection 4 OMI L1B radiance and irradiance data are available, free of charge, at Goddard Earth Sciences Data and Information Services Center (https://disc.gsfc.nasa.gov/datasets/OML1BRUG_004/summary). The experimental OMI PCA $SO_2$ SCDs and NN analysed SCDs are available upon request from the corresponding author. Code used to analyse data and produce figures in this paper is also available upon request from the corresponding author.

## Author contribution

CL and JJ designed the NN-based analysis method. CL implemented the method, performed tests, and prepared the manuscript. JJ provided the code and NN architecture used in the study. FL conducted top-down emission estimates. VF and CM designed and provided the emission algorithm. All authors commented on the manuscript.

## Competing interests

The authors declare that they have no conflict of interest.

## Acknowledgements

We would like to thank Dr. Arlindo da Silva of NASA Goddard Space Flight Center for comments and suggestions on the interpretation of the NN analysed results. We also thank the NASA Earth Science Division (ESD) Aura Science Team program for funding of the OMI $SO_2$ product development and analysis. The Ozone Monitoring Instrument (OMI) is a Dutch/Finnish instrument flying aboard the NASA Earth Observing System Aura spacecraft. The OMI project is managed by the Royal Meteorological Institute of the Netherlands (KNMI) and the Netherlands Space Agency (NSO).

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

**Table 1.** The correlation coefficent ($r$) and root mean square error (RMSE) between the NN analysed OMI $SO_2$ SCDs ($SCD_{NN}$) using different predictors and the target SCDs ($SCD_{target}$). The NNs are trained using data from July 5, 10, 15, 20 and 25, 2005. The comparisons shown here are for pixels from the same days but not included in the training.

| Predictors | $r$ | RMSE (DU) |
|---|---|---|
| $SRR_i$ | 0.642* | 0.180* |
| $SRR_i + SRR_m$ | 0.958 | 0.0517 |
| $SRR_i + SRR_m + \theta$ | 0.962 | 0.0491 |
| $SRR_i + SRR_m + R + \theta$ | 0.968 | 0.0451 |
| $SRR_i + SRR_m + R + \theta_0 + \theta$ | 0.976 | 0.0393 |
| $SRR_i + SRR_m + R + \theta_0 + \theta + \phi$ | 0.976 | 0.0392 |
| $SRR_m + R + \theta_0 + \theta + \phi + O_3$ | 0.793 | 0.111 |
| $SRR_i + R + \theta_0 + \theta + \phi + O_3$ | 0.978 | 0.0374 |
| $SRR_i + SRR_m + R + \theta_0 + \theta + \phi + O_3$ | 0.976 | 0.0388 |

*Results shown are from a simple linear regression anlysis.

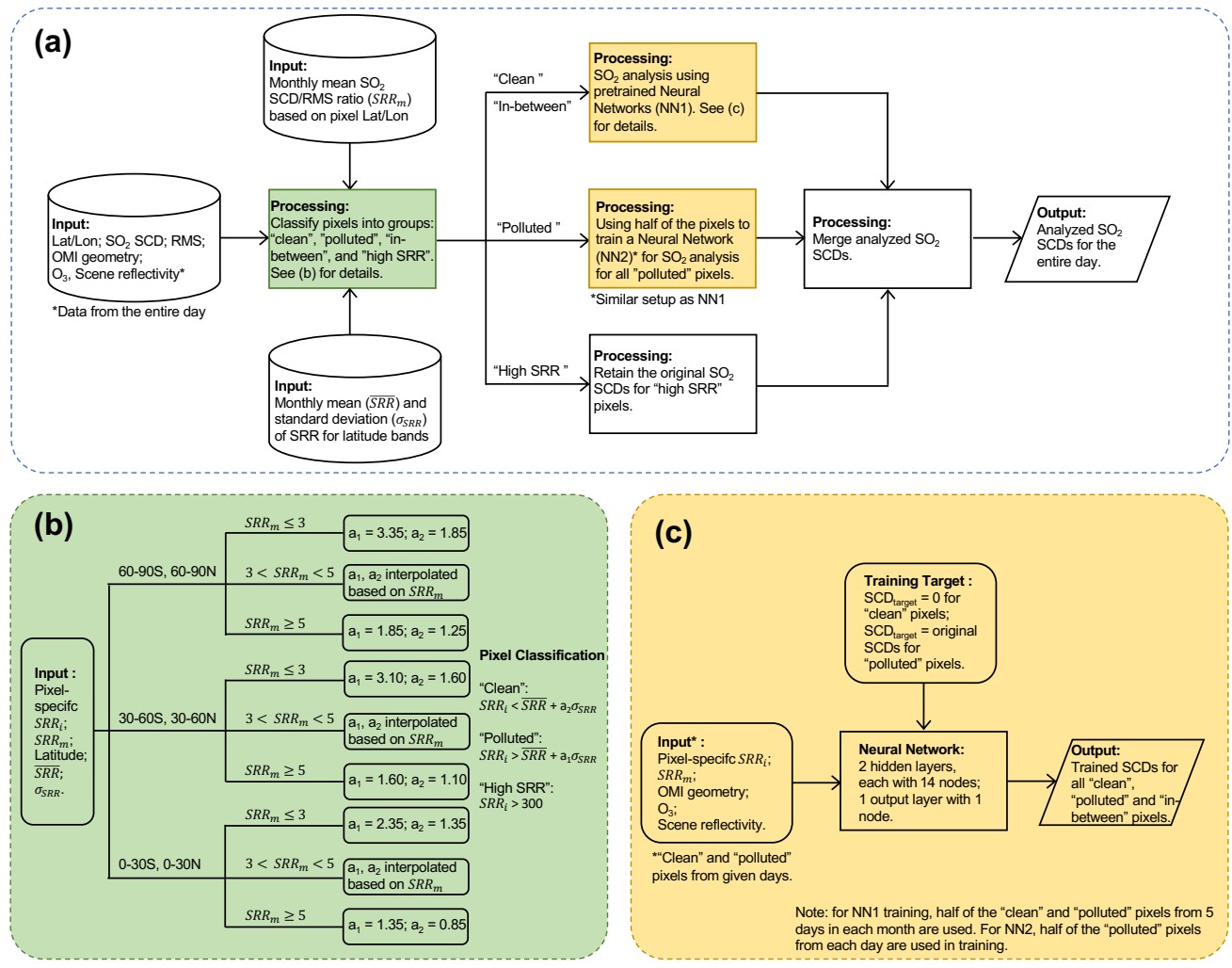

**Figure 1: (a)** Flow chart of the SO₂ analysis method. **(b)** Scheme for classification of OMI pixels as "clean", "polluted", "in-between" and "high-SRR". **(c)** Setups of the neural networks for SO₂ SCD analysis.

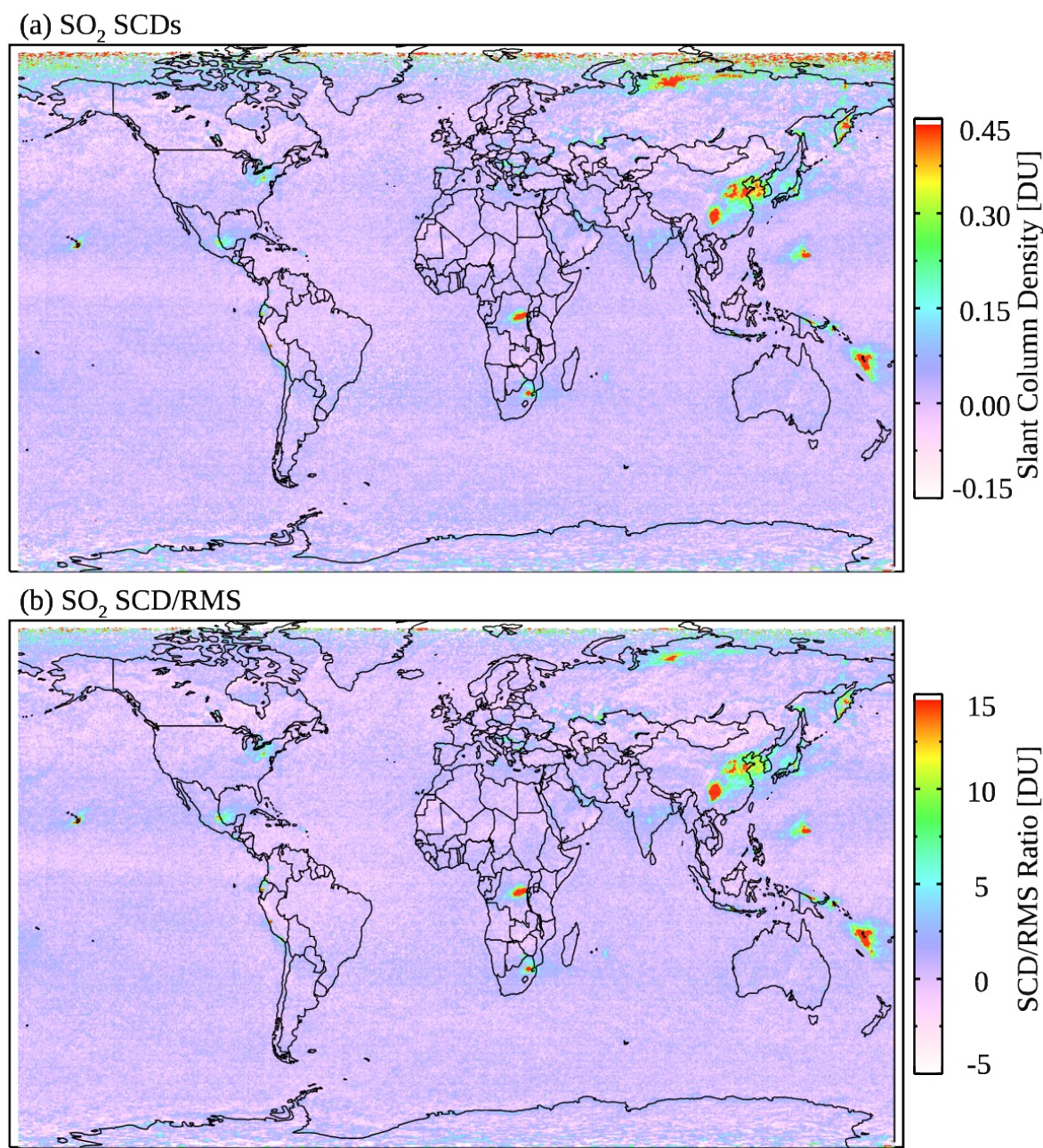

**Figure 2: (a) Monthly mean OMI SO$_2$ SCDs for March 2005 showing enhanced SO$_2$ signals over major anthropogenic source areas (e.g., China, the eastern U.S., India, and South America) as well as degassing volcanoes. Note the positive bias at northern high latitudes. (b) Monthly mean SCD/RMS ratio (SRR) from the same sample of OMI pixels as in (a). The SRR map also shows major SO$_2$ sources but has reduced bias at high latitudes as compared with the SCD map.**

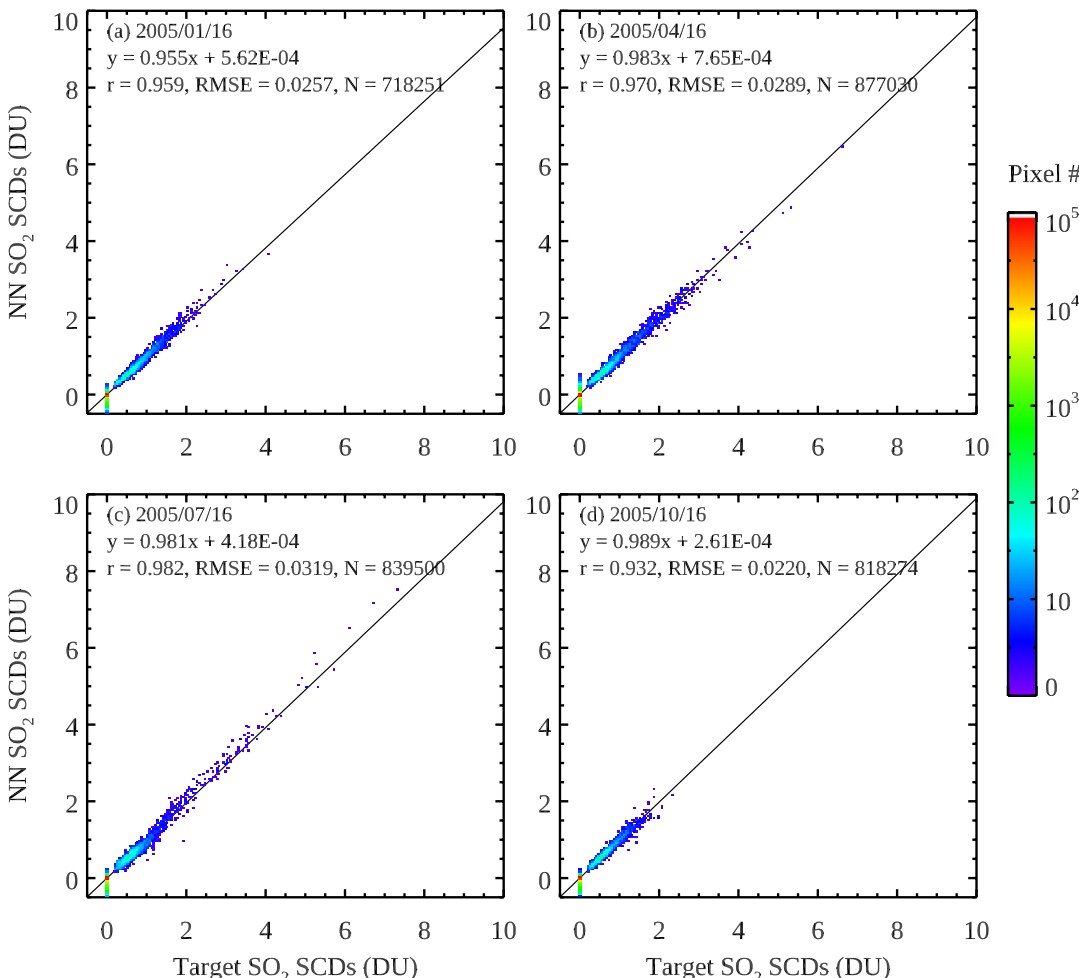

**Figure 3: Scatter plots between the NN analysed SO₂ SCDs and the target SO₂ SCDs for clean and polluted OMI pixels from the 16th day of (a) March, (b) April, (c) July, and (d) October 2005. Only pixels not used in the training of the neural networks are shown. Colours represent the number of data points within each 0.1 DU (in NN SCDs) by 0.1 DU (in target SCDs) bin. The solid line in each panel represents the best fit through the data from the simple linear regression analysis between NN and target SCDs. The slope and intercept from the regression are given in each panel, along with the correlation coefficient (r), root mean square error (RMSE), and number of pixels (N).**

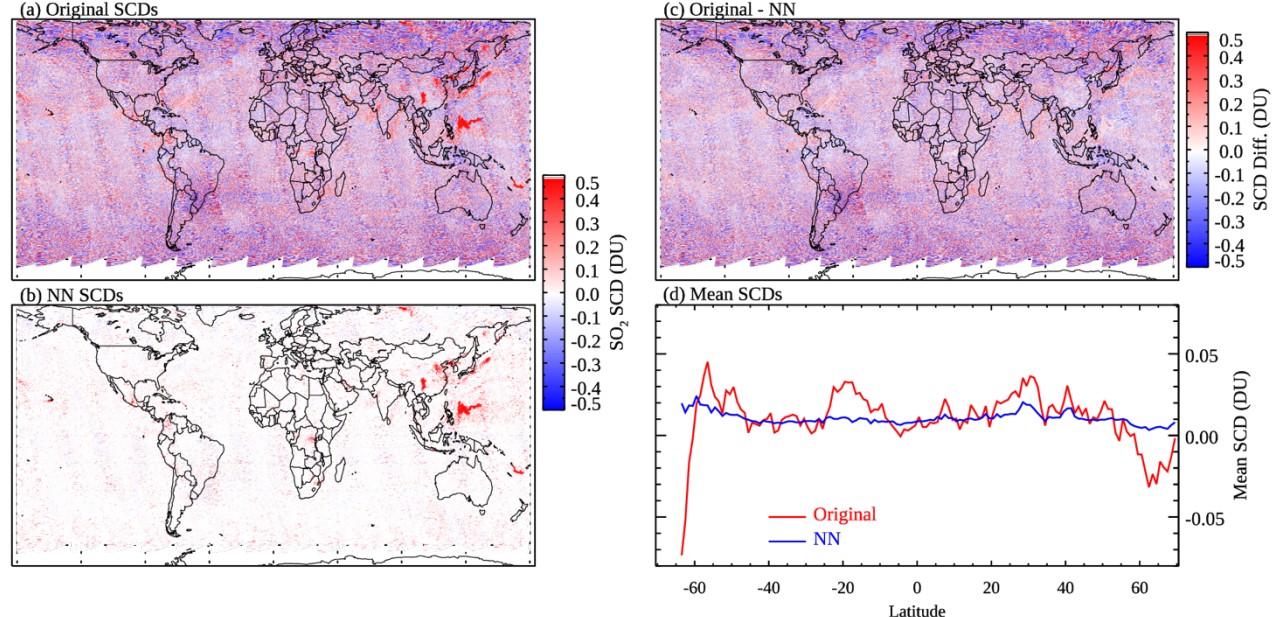

**Figure 4:** The (a) original, (b) NN analysed OMI SO₂ SCDs and (c) their differences for April 16, 2005. (d) Mean SO₂ SCDs for 1° latitude bands over generally clean areas (monthly mean SRR < 3), calculated from (red) the original and (blue) NN analysed SCDs for the same day.

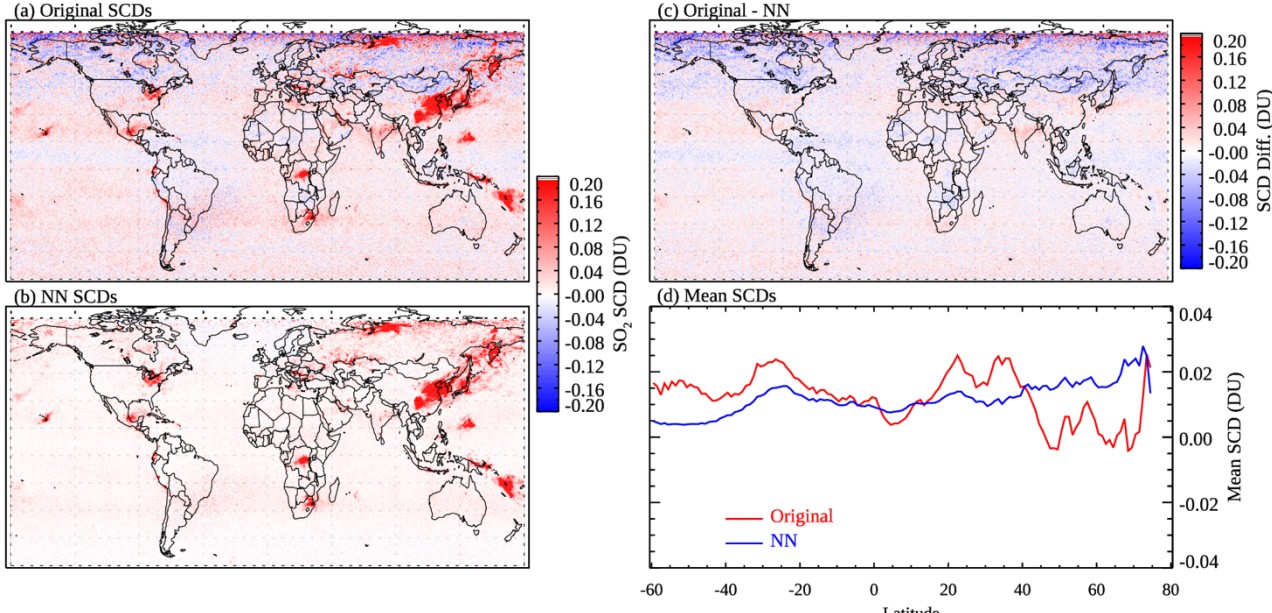

**Figure 5: Similar to Figure 4 but showing monthly means for March 2005.**

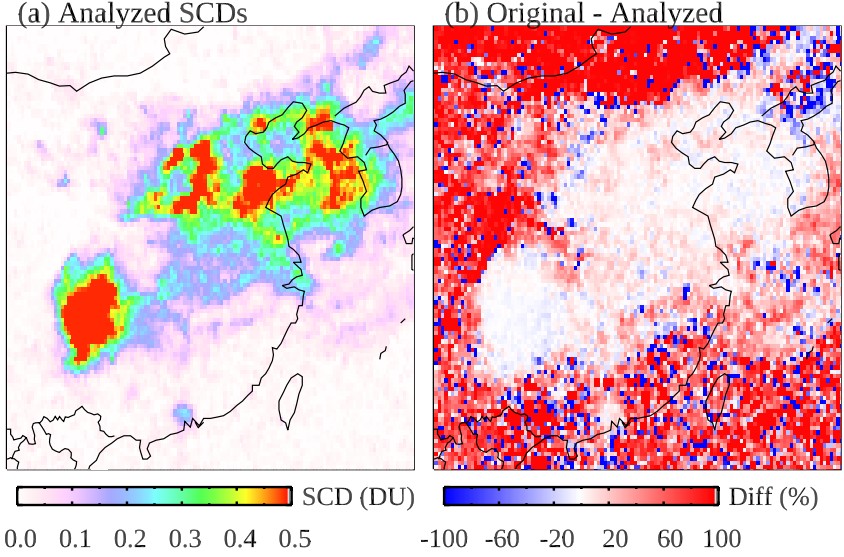

**Figure 6: (a) The NN analysed SO₂ SCDs and (b) their relative differences from the original SCDs over eastern China for March 2005.**

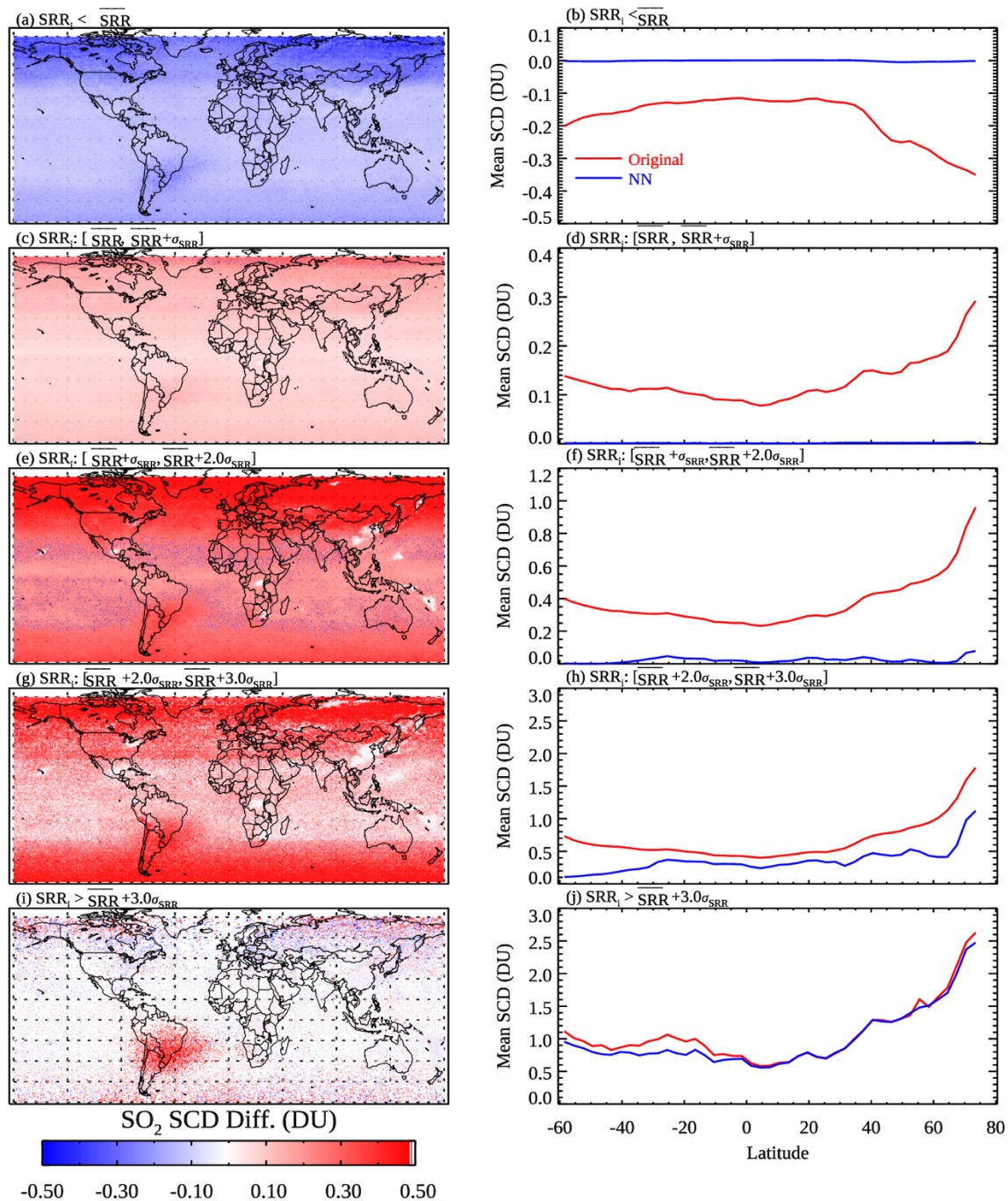

**Figure 7: Left:** the differences between the original and the NN analysed OMI SO₂ SCDs for March 2005. **Right:** the mean (red) original and (blue) NN analysed SCDs for 3º latitude bands for the same month. Different rows show results from pixels that have SCD/RMS ratios (SRR$_i$) within different ranges based on the monthly medians of the daily mean ($\overline{SRR}$) and standard deviation ($\sigma_{SRR}$) of SRRs for their corresponding latitude bands: (a-b) SRR$_i$ < $\overline{SRR}$, (c-d) $\overline{SRR}$ < SRR$_i$ < $\overline{SRR}$ + $\sigma_{SRR}$, (e-f) $\overline{SRR}$ + $\sigma_{SRR}$ < SRR$_i$ < $\overline{SRR}$ + $2\sigma_{SRR}$, (g-h) $\overline{SRR}$ + $2\sigma_{SRR}$ < SRR$_i$ < $\overline{SRR}$ + $3\sigma_{SRR}$ and (i-j) SRR$_i$ > $\overline{SRR}$ + $3\sigma_{SRR}$.

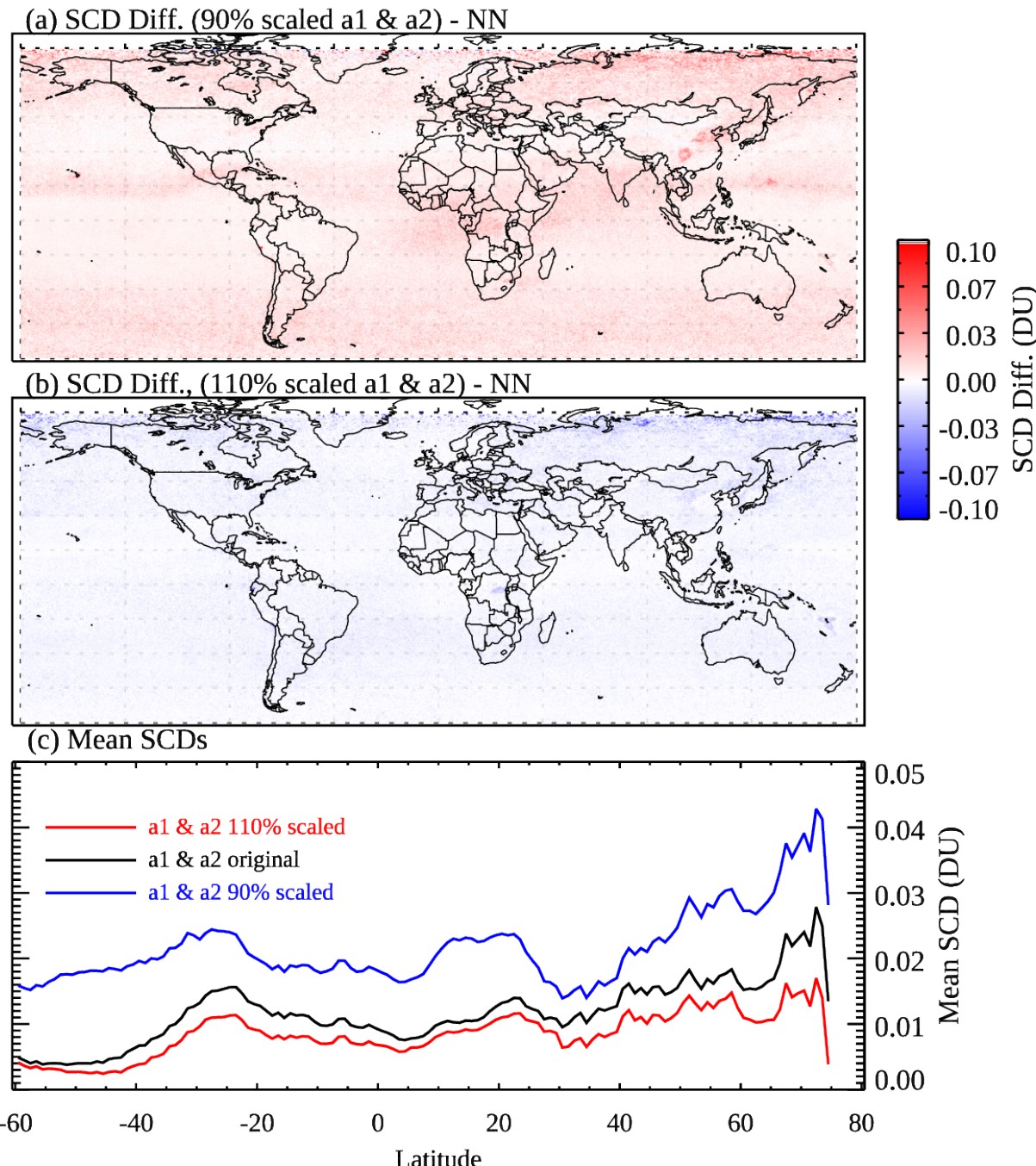

**Figure 8: (a) Differences in the analysed OMI SO$_2$ SCDs for March 2005 between NNs trained using pixels classified with a$_1$ and a$_2$ (see Eqs. 1 and 2) scaled to 90% of the baseline values in the classification scheme and those trained with the baseline scheme. (b) Same as (a) but for a$_1$ and a$_2$ scaled to 110% of the baseline values. (c) Mean SCDs for 1° latitude bands over relatively clean areas (monthly mean SRR < 3) using NNs trained with pixels from different classification schemes: (black) the baseline a$_1$ and a$_2$, (blue) a$_1$ and a$_2$ scaled to 90% and (red) a$_1$ and a$_2$ scaled to 110% of the baseline values.**

625

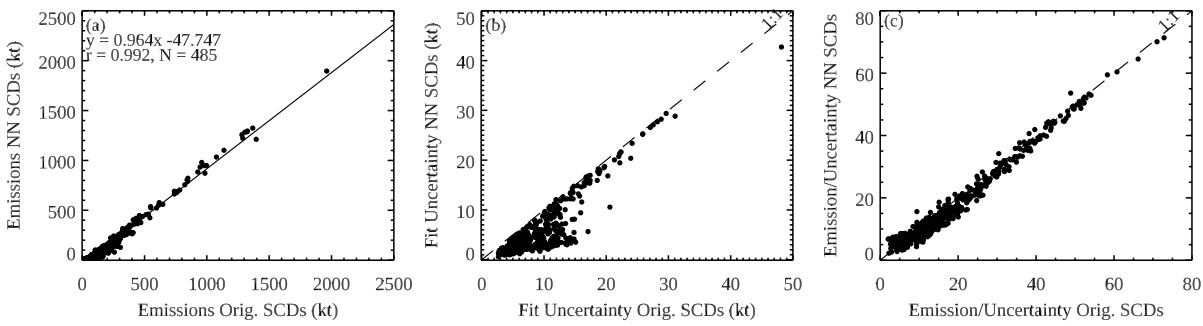

**Figure 9: Scatter plots comparing (a) the annual emission estimates for 485 large point sources for 2005, (b) the uncertainties in the emission estimates, and (c) the ratios between the emission estimates and the uncertainties using the NN analysed *vs.* the original SCDs. All sources shown here are anthropogenic and have emission estimates at least twice the uncertainties for both datasets.**

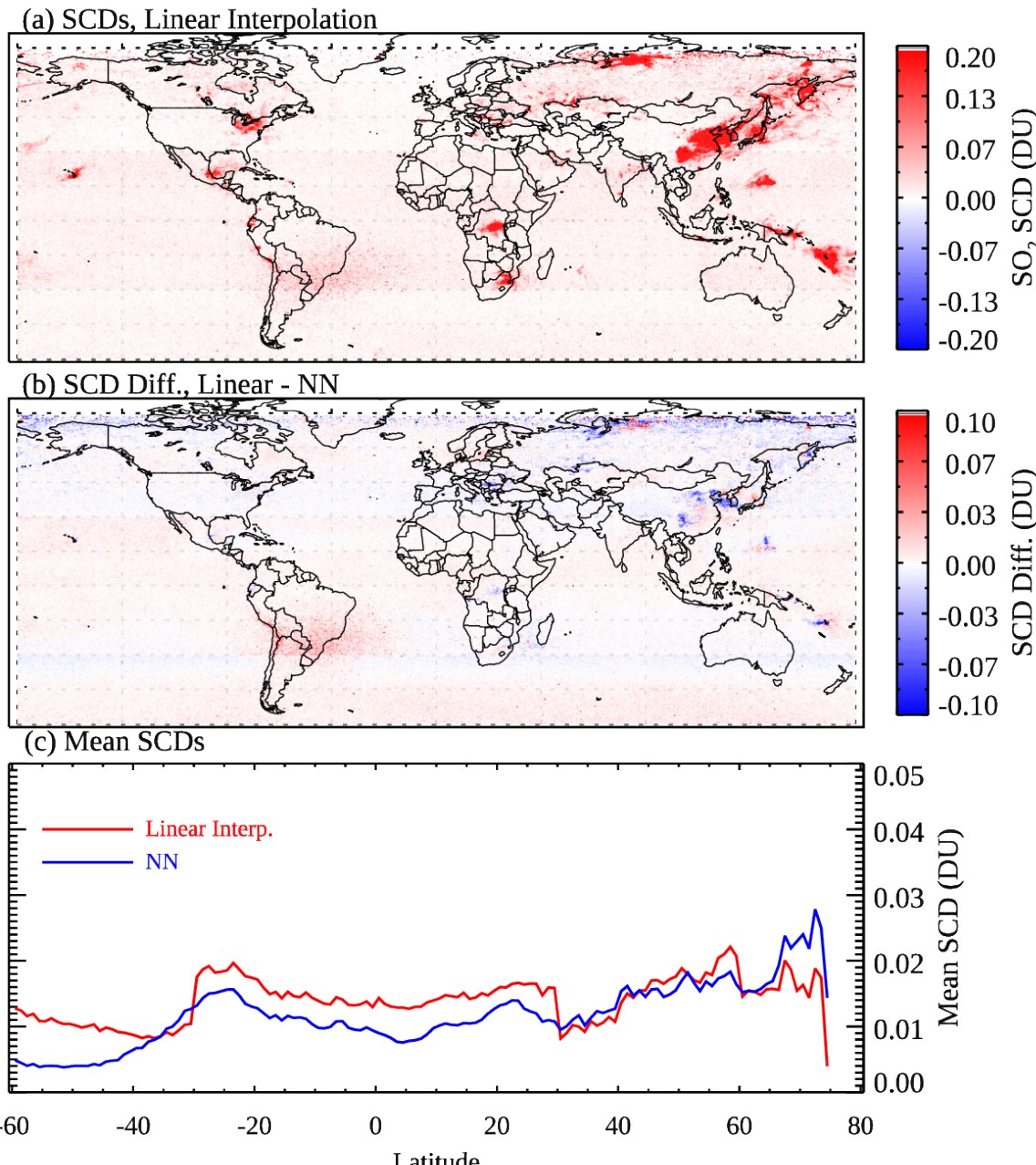

**Figure 10: (a)** Monthly mean OMI SO₂ SCDs for March 2005, analysed using a simple linear interpolation model. **(b)** The differences in the analysed SCDs between the linear model and the neural networks. **(c)** Mean SCDs for 1° latitude bands over generally clean areas (monthly mean SRR < 3), calculated from the SCDs from (red) the linear interpolation model and (blue) the NNs.

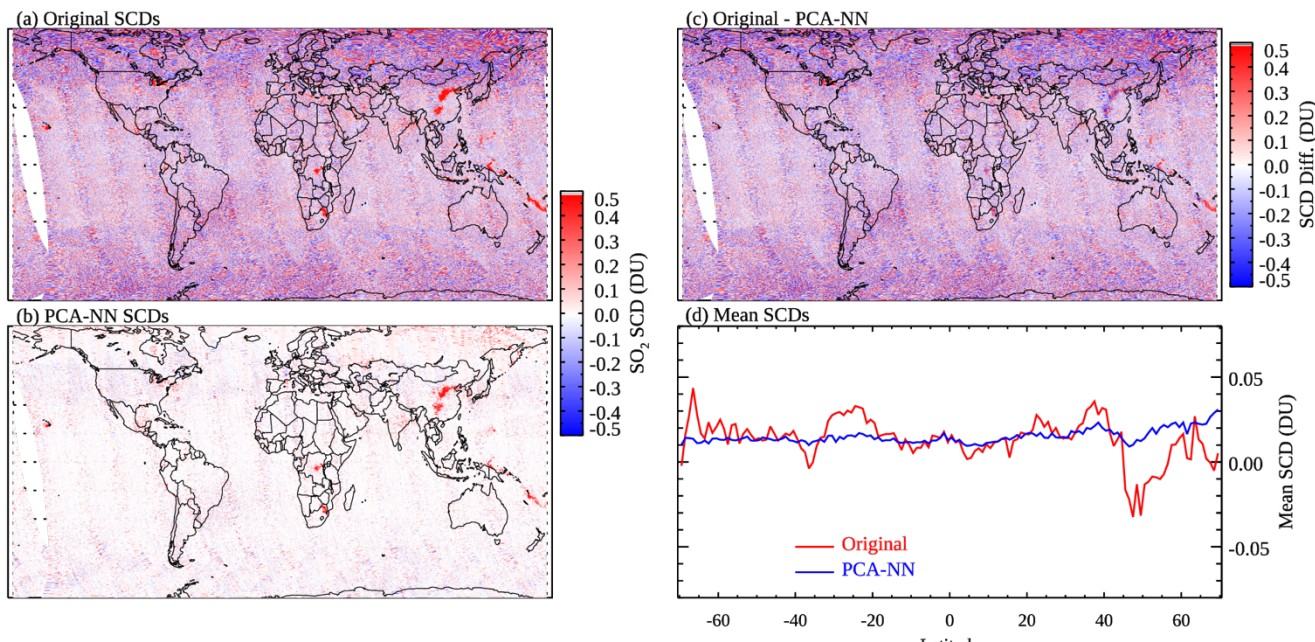

**Figure 11: OMI SO₂ SCDs for April 16, 2005 retrieved using (a) the original PCA algorithm and (b) a PCA-NN algorithm, (c) the differences between the two retrievals, and (d) mean SO₂ SCDs for 1° latitude bands over relatively clean areas (monthly mean SRR < 3), calculated from (red) the original and (blue) PCA-NN retrievals.**

.