# Peer review of "A New Machine Learning based Analysis for Improving Satellite Retrieved Atmospheric Composition Data: OMI SO2 as an Example"

_Atmospheric Measurement Techniques, 2022_

## Author Comment (AC1)

**Response to Referee Comment 1 (RC1) from Anonymous Referee # 2** (*Referee comment in Italic*, response in blue).

**RC1**: 'Comment on amt-2022-135', Anonymous Referee #2
*Li et al. presented an new machine learning scheme and applied it to OMI SO2. It is an interesting preliminary study and the work should be published after minor revision. The paper is well written and structured. Overall, the figures are not of sufficient resolution.*

We thank the referee for the review and for raising several important points. Below please find our point-to-point response. As for the figures, we have made changes to Figure 7 to make it more readable. We will also provide high-resolution files to AMT for the final production, if our paper is accepted.

***Main comment:*** *although it is an interesting study, I am concerned by the fact that the NN function is heavily weighted by SRR. The SRR, as defined by the author, actually contains the desired result. Therefore, it is not a surprise that this works. The only real task of the NN is to resolve the RMS dependence. Also, the fact that the noise gets reduced is in a way artificial, as it is the direct result of the constrain on the clean pixels that should be SCD = zero.  The reduction of the bias poses also the problem of a possible overcorrection. On long-term averages, are the weak emissions sources detected by OMI PCA still present in the NN data set?*

*In my opinion, what would be much stronger is to train the NN directly with the corresponding radiances. This is done to some extend (through a PCA transformation) but it is coming at the end of the paper. It is  pity it is not put more in front.*

We agree that the NNs are heavily weighted by SRR. At the same time, based on the correlation coefficient and RMSE in Table 1, a model based on just SRR will produce fairly large errors. Also as shown in sect. 4.3 and Figure 9, a more complex model based on SRR and the additional spatial context (monthly mean SRR) still underestimates $SO_2$ over polluted areas. In that sense, we feel that the NN and the additional variables have certainly helped to improve the results.

We also agree that assigning zero SCDs to clean pixels is a fairly strong constraint, which is to some extent out of necessity given the dearth of measurements over vast remote regions. In a way, our analysis method can be viewed as a more advanced version of the Pacific sector correction (PSC), a quite common and well-established practice to reduce retrieval artifacts for species such as $SO_2$ and HCHO. In the PSC approach, retrievals over the remote Pacific are averaged as a function of latitude. The latitude-dependent mean VCDs or SCDs (or the differences between retrievals and model simulations) over the remote areas are considered artifacts and subtracted from retrievals for all pixels, regardless of their location. Our approach, on the other hand, considers more factors for each individua pixel. We have added this point to the revised paper.

As for the overcorrection, we have taken caution (mainly through the test of different setups for the pixel classification scheme) to reduce it. The test on the annual emission estimates in the paper shows

that the weaker sources are still present in the annually averaged NN analyzed data. We have added the potential overcorrection as a caveat in the revised paper.

Finally, we agree that training the NNs to do retrievals from radiances would be interesting and valuable. But at this point there is still a lot of room for improvement as clearly shown in sect. 4.4 and Figure 10 (sect. 4.5 and Figure 11 in the revised paper). We plan to conduct follow-up studies to explore ways to further improve the NN-based retrievals from radiances. Also the lack of high-quality training data is a major obstacle in training NNs for retrievals. And in this sense, our current study contributes to such efforts, by providing training data with improved quality (as compared with the original retrievals). We have added this point to the revised paper.

**Minor comments**

*L55: I agree with the necessity to improve the retrievals but is a 10% noise increase a real problem for addressing long-term trend monitoring? I do not think so. The appearance of instrument issue like row anomaly is more of a problem.*

The point we try to make here is that with reduced emissions (and signals), the increase in noise makes the long-term monitoring more challenging, especially if the signal is relatively weak to begin with. We have clarified this point in the revised text.

*Figure 2b is confusing. Is the SRR unit less?  The SCD is DU and RMS has no unit, thus SRR should be expressed in DU (?).*

Thank you for pointing this out. We have made the correction to Figure 2b in the revised paper.

*Section 2.2. The classification of pixels is quite complex. As explained in the text, the parameters used (a1,a2) have been determined by testing. However, it would be good to illustrate the impact of the (a1,a2) settings on the final results. Currently, it is hard to judge if the complexity is worth, compared to a more simple classification.*

We started from a simpler classification scheme (with fixed a1 and a2) but found some deficiencies. For example, we found signs of overcorrection (negative bias) for sources at lower latitudes. We also found relatively large positive bias for high latitudes. This is why we moved to the more complex scheme. We have added this discussion to the revised paper.

Per your suggestion and also the comment from Referee #1, we have conducted additional sensitivity tests by altering a1 and a2 by +/-10%. Overall, the final results show some (but not overly large) sensitivity to the a1 and a2 parameters. We have added these test results and the discussion to the revised paper.

*Section 2.3: the processing is not performed separately for each row. Why not? Would it improve/degrade the results?*

The main reason is to ensure a large enough sample of polluted pixels in the training data. On average, each row would only have ~200 polluted pixels per day. In the PCA-based retrievals, we process each row separately because each has different measurement characteristics (e.g., wavelengths, instrument spectral response function). After the PCA SCD retrievals, the row-to-row difference in SCDs and SRRs are relatively small (see figure below). And we do not expect major changes in the training results if we process the data separately for each row.

[Figure]

Figure. Mean $SO_2$ SCDs and SRRs for each of the 60 OMI rows for April 16, 2005, calculated from pixels with SZA < 70, outside of the SAA areas, and outside of polluted regions (monthly mean SRR < 3).

*Section 3.1: the SCDs over SAA are much better in the NN analysis, which is surprising. Any idea why?*

One potential reason is that retrievals over SAA areas tend to have relatively large fitting residuals (and RMS). The use of SRR partially cancels out the relatively noisy SCDs. We have added this point to the revised paper.

*Figure 7: the figure quality is not sufficient. When zooming over the subplots it is hard to see the emission patterns described in the text.*

Per your comment and also the suggestion from Referee #1, we have revised Figure 7 to include fewer subplots. We have also moved some subplots (as separate figures for each SRR range) to the supplemental information. The text has also been updated accordingly.

*Figure 10b : the PCA-NN results seem to show striping features, although the analysis is performed separately for each row. Why?*

The NN models are trained separately for each row but using the same architecture, and the training performance varies for different rows. The reason for this varying performance is unknown to us at this point and will be the subject of further investigations. The training performance may improve if the architecture is optimized for each row, but this will require substantial effort and is probably more suitable for a follow-up study.

---

## Author Comment (AC2)

**Response to Referee Comment 2 (RC2) from Anonymous Referee # 1** (*Referee comment in Italic*, response in blue).

***Review of manuscript "A New Machine Learning based Analysis for Improving Satellite Retrieved Atmospheric Composition Data: OMI SO2 as an Example" by Can Li et al. 2022***

*In the manuscript a new and interesting method to minimize noise and artifacts using machine learning has been presented and applied to OMI SO2 data. The paper is well written and only minor revisions are needed.*

*My **main comment** is that an optimization (or sensitivity analysis) of the applied NN architecture is missing. A simple NN architecture from a completely different scientific field ("for reconstruction of RGB images from hyperspectral radiances") was used. In general a simple architecure is fine as a starting point, but what I am clearly missing is a sensitivity analysis of much the NN architecture is affecting the results.*

*Furthermore I suggest some restructuring of the paper - I think it would be good to start with the simple approach of using the linear interpolation model and then begin the NN model.*

*Althoug the comparison and different maps are nice to see, I suggest to also add line plots as a function e.g. latitude such that is easier to see biases and differences.*

We thank the referee for providing several helpful suggestions. The NN architecture used in this study has also been applied to a few other studies on different topics and we have added references to those. Additionally, we have also conducted more tests on the architecture, please see below our response to the specific suggestion.

We have also carefully considered the suggestion to move the part of the paper on the linear interpolation model to the front. We agree that this change could help to justify the use of NN in this study. A potential issue is that the linear model results are compared with NN analyzed results, so it would be necessary to introduce the NN approach first. Because of this, we have elected to keep the overall structure of the paper unchanged. We have added, in the methodology section, that a simple linear interpolation model does not work quite as well as the NN based approach.

We have also added line plots to several figures following the suggestions.

**Detailed comments:**

- *Sect.1, L49: Please provide numbers or references for the background noise of OMI SO2 SCDs.*

   Reference added.

- *Sect 1. L80ff: The 20 DU limit of the FP-ILM retrieval only applies to the SO2 LH retrievals. It was not yet applied to SO2 VCD retrievals.*

We have clarified this point in the revised paper.

- *Sect 2.1, L109: How are pixels with enhanced SO2 after volcanic eruptions detected/filtered? Do you apply a VCD threshold? Please describe.*

In the PCA retrieval algorithm, this is done by examining the $O_3$ residuals at two wavelength pairs (313/314 nm and 314/315 nm). We have added the reference for this in the revised manuscript.

- *Sect 2.1, L128: By how much do the monthly medians/stddev change every month? Do you see jumps in the results from one month to the other?*

The monthly medians and standard deviation are strongly affected by SZA. When calculated as a function of latitude, both show seasonal changes (please see figure below ).

[Figure]

Figure. (Left) Monthly median and (right) standard deviation of SCD/RMS (SRR) ratios for different latitude bands for January and July 2005.

- *Sect. 2.2, L146: I am a bit concerned that the a1/a2 factors are based on trial and error and there is no robust criterion to determine them. This makes appying the whole method to other sensors (or even to OMI for a different time span) difficult - I assume the factors will vary over time, especially related to instrument degradation. Can you perhaps show how sensitive your results are to (small) changes in a1/a2?*

Following your suggestion and also that from the Referee #2, we have conducted sensitivity tests by changing a1 and a2 by +/-10%. While the NN analyzed SCDs show some sensitivity to a1 and a2, overall the sensitivity is not very large ($< 0.1$ DU for polluted areas and 0.03 DU or less for clean areas). With that, we think it would probably be better to use a constant set of a1 and a2 parameters for long-term data analysis. For applications to other instruments or datasets, a new set of parameters are most likely

needed given the differences between instruments/retrievals. We have added these results and the discussion to the revised paper.

• *Sect 2.3, L180ff: As already mentioned in my main comment, I am a bit concerned about the choice of the NN architecture. Although it is a good starting point to use a simple archictecture, it definitely needs to be optimized for the specific problem, especially independently for your NN1 and NN2.*

*Is there a reason for the choice of activation functions? I.e. using soft-sign and then sigmoid is not really common - I suggest to use ReLu or something related for both hidden layers.*

Thank you for the suggestion. We have conducted some additional tests. In one, we altered the number of nodes in the hidden layers (to 20 for both layers and then to 30). We found only marginal changes in the performance of the NNs. For example, for training for March 2005, the RMSE of NN SCDs are 0.0331 DU, 0.0330 DU, and 0.0329 DU for our current setup, 20 nodes per hidden layer, and 30 nodes per hidden layer, respectively.

In another test, we used ReLU as the activation function for both hidden layers. We notice that the new setup speeds up the training and reduces the RMSE of NN SCDs from 0.0331 DU to 0.0319 DU for March 2005. But once we applied the trained NNs to the entire month, we notice that the new setup overall increases the SCDs over background areas by ~0.01-0.02 DU (see figure below). Of course, the pixel selection scheme has been optimized for our existing setup, so this may not be a completely fair comparison. We have added this discussion to the revised manuscript (with the figure added to the supplemental information) and we intend to conduct more extensive tests in follow-up studies.

[Figure]

Figure. (a) Monthly mean analyzed SCDs for March 2005 using ReLU as the activation function in both hidden layers of the NNs. (b) The differences in the monthly mean analyzed SCDs for the same month between the NNs using ReLU as the activation function and the original NNs (soft sign and sigmoid as the activation function for the first and second hidden layer, respectively). (c) The mean $SO_2$ SCDs for 1-degree latitude bands from the two NN architectures over generally clean areas (monthly mean SRR < 3).

- *Sect. 4.3: I suggest to put this in front before you apply the NNs, to show that a simple linear interpolation is not sufficient and therefore you apply NNs.*

As mentioned above, the linear model results are compared with NN results in this section, and it is therefore necessary to introduce NN results first. We feel that it is probably better to keep the section in its current place.

- *Figure 4&5: Suggest to add line plot of SCD as a function of latitude. With this plot you probably better see biases and the differences*

Line plots added.

- *Figure 5: Suggest to use different color scale from -0.1 - 0.1*

We experimented with a few different color scales and selected -0.2 to 0.2 DU in the updated figure.

- *Figure6b: Relative differences are always problematic for SO2 plot since in clean areas the SCD is close to zero and hence the relative difference becomes extremly high, as can be seen in the figure. Suggest to use absolute differences here.*

We did try to plot absolute differences instead of relative differences for Figure 6 (please see below). We feel that it is more difficult to distinguish between polluted and clean areas from the absolute differences. For this reason, we have elected to keep the relative difference plot in Figure 6.

[Figure]

Figure. Same as Figure 6 but showing absolute differences in (b).

- *Figure 7: The maps are rather confusing and do not provide addtional information. I sgueest to rather repleace them by line plots as a function of e.g. latitude (see my comment above)*

Following your suggestion and also that from Referee #2, we have changed Figure 7 by adding line plots and also moving some subplots (maps) to the supplemental information. These changes make Figure 7 more readable and easier to interpret. We have also updated the text in the revised paper accordingly.

- *Figure 9&10: Suggest to add line plots of SCD as a function of e.g. latitude. With this plot you probably better see biases and the differences.*

Line plots added.